# Sentiment Classification Method Based on Blending of Emoticons and Short Texts

**DOI:** 10.3390/e24030398

**Published:** 2022-03-12

**Authors:** Haochen Zou, Kun Xiang

**Affiliations:** 1Department of Computer Science and Software Engineering, Concordia University, Montreal, QC H3G 1M8, Canada; 2Department of Science and Engineering, Hosei University, Koganei 184-8584, Tokyo, Japan; kun.xiang.2u@stu.hosei.ac.jp

**Keywords:** sentiment analysis, convolutional neural network, emoticon vectorization algorithm

## Abstract

With the development of Internet technology, short texts have gradually become the main medium for people to obtain information and communicate. Short text reduces the threshold of information production and reading by virtue of its short length, which is in line with the trend of fragmented reading in the context of the current fast-paced life. In addition, short texts contain emojis to make the communication immersive. However, short-text content means it contains relatively little information, which is not conducive to the analysis of sentiment characteristics. Therefore, this paper proposes a sentiment classification method based on the blending of emoticons and short-text content. Emoticons and short-text content are transformed into vectors, and the corresponding word vector and emoticon vector are connected into a sentencing matrix in turn. The sentence matrix is input into a convolution neural network classification model for classification. The results indicate that, compared with existing methods, the proposed method improves the accuracy of analysis.

## 1. Introduction

As an important media platform for spreading social events, the Internet plays a significant role in social events [1]. With the rapid development and maturity of Internet technology, many online social platforms have gradually become the main media for people to obtain information and communicate with each other. Twitter as a social network platform is popular because of its real-time, convenient, and interactive characteristics [2]. The burgeoning increase of Twitter and other social platforms depends on the following two points. First, the short length of tweet text reduces the threshold of information production and reading, catering to the trend of fragmented reading in the current fast-paced life [3]. Second, social network content such as tweets can contain texts, emojis, pictures, videos, and other forms, which makes up for the lack of pure text communication compared with face-to-face communication and makes text communication more immersive and accurate [4]. Users can log in to Twitter and publish information by computers, smartphones, and other terminal devices. There are two striking features of short texts such as tweets. First, they are short, together with a word limitation on tweets [5]. The short content of the tweet means it contains relatively narrow information, which is not conducive to further analysis. Second, short texts such as tweets and comment content contain a wealth of emojis [6]. Emojis have been used frequently on social media, and they have been endowed with rich connotations in the process of use. In addition to basic functions such as expressing actions (e.g., “Go skiing today! 
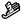
”), objects (e.g., “Sushi 
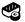
 for lunch.”), weather (e.g., “It’s sunny 
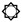
 in Montreal this morning.”) or emotions (e.g., “Tonight is a great night! 
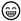
”), emojis can also enhance the emotion and even disambiguate short texts such as tweets with sarcastic words and phrases (e.g., “I love to work overtime! 
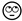
”).

The additional content, such as pictures, videos, and links, has little influence on the emotional inclination of the tweet itself, which is a kind of noise and can be eliminated from study. Tweet short text, as one of the most important elements of Twitter, determines the emotional orientation of tweets in most cases [7]. However, due to the limited number of words, short text sometimes cannot fully express users’ emotions and attitudes. Therefore, users add emoticons such as emojis to enrich their emotional leanings. Internet emoticons were born in the 1980s, and the original emoticons were made up of characters [8]. With the advancement of the Internet, emoticons have undergone great changes in form, content, and function. Emojis have moderately become the most popular emoticons on social networks and have become an indispensable chatting tool in today’s network communication [9]. Symbolic communication can convey feelings more accurately and change people’s communication mode and expression habits. At the same time, emoticons have different extended meanings in different situations [10], which can convey rich semantic information beyond the reach of text expression. Therefore, the importance of emoticons is self-evident. To objectively judge the emotional polarity of short texts such as tweets, it is necessary to study emoticons such as emojis in addition to analyzing short text. By integrating emoticons into the process of short-text sentiment analysis, it can more accurately judge the emotional tendency of short texts in social media such as Twitter and TikTok.

Social networks are not only a medium for people to record their lives and communicate with each other but also a way to express personal feelings and maintain relationships [11]. Therefore, social media such as Twitter is an important carrier for people to express their happiness and sorrow. As an extremely influential news and public opinion platform, short texts from social networks generate huge emotional information from a great number of users, which seems to be chaotic but contains a considerable value. These emotional traits reflect users’ interests and preferences and, at the same time, may also have a huge impact on the spread of online public opinion [12]. Therefore, sentiment analysis of short texts can understand users’ preferences and their views on some hot events in real society and make trend predictions, providing the scientific basis for government decision making. At the same time, tweets and other short-text data from social media contain a vast majority of users’ comments and suggestions on products, services, environment, etc. [13]. Enterprises and institutions can further mine and analyze short-text information to obtain and provide a scientific basis for further research and research or improvement of products [14]. Through the analysis of short-text information, we can not only predict people’s personality characteristics and living conditions but also forecast the development trend of new events, which has practical significance for social development.

Previous work has studied the presence of sentiment value in different short texts and attempted to analyze the relevant sentiment characteristics in these cases. Zhao et al. proposed an unsupervised word-embedding method based on large corpora, which utilizes latent contextual semantic relationships and co-occurrence statistical features of words in tweets to form effective feature sets. The feature set is integrated into a deep convolution neural network for training and predicting emotion classification labels [15]. Alharbi et al. proposed a neural network model to integrate user behavior information into a given document and evaluate the data set using a convolutional neural network to analyze sentiment value in short text such as tweets. The proposed model is superior to existing baseline models, including naïve Bayes and support vector machines for sentiment analysis [16]. Sailunaz et al. incorporated tweet responses into datasets and measurements and created a dataset with text, user, emotion, sentiment information, etc. The dataset was used to detect sentiment and emotion from tweets and their replies and measured the influence scores of users based on various user-based and tweet-based parameters. They used the latter information to generate generalized and personalized recommendations for users based on their Twitter activity [13]. Naseem et al. proposed a transformer-based sentiment analysis method, which encodes the representation from the converter and applies deep intelligent context embedding to improve the quality of tweets by removing noise and considering word emotion, polysemous, syntactic, and semantic knowledge. They also designed a two-way long-term and short-term memory network to determine the sentiment value of tweets [17]. However, these studies focus on methods to improve the accuracy of analyzing the emotional characteristics of short texts, ignoring the effect of emoticons such as emojis on the emotional tendency of the whole text.

Several studies have been conducted to analyze the emotional features and semantic information contained in emoticons and their emotional impact on text content. Barbieri et al. collected more than 100 million tweets to form a large corpus, and distributed representations of emoticons were obtained using the skip-gram model. Qualitative analysis showed that the model can capture the semantic information of emoticons [18]. Kimura et al. proposed a method of automatically constructing an emoticon dictionary with any emotion category. The emotion words are extracted, and the co-occurrence frequency of emotion words and emoticons is calculated. According to the proportion of the occurrence times of each emoticon in the emotion category, a multi-dimensional vector is assigned to each emoticon, and the elements of the vector represent the intensity of the corresponding emotion [19]. However, these studies focused on the emotional characteristics and semantic information contained in emoticons themselves, rather than combining emoticons with textual data. Arifiyanti et al. utilized emoticons to build a model, classified the emotion categories of tweets containing emoticons, and evaluated the performance of the classification model [20]. Helen et al. proposed a method to understand emotions based on emoticons, and a classification model based on attentional LSTM was designed [21]. However, the above studies focus on understanding the emotional characteristics of the whole text by extracting and analyzing emoticons in the text and establishing relevant sentiment labels, without integrating emoticons’ emotional information with the emotional features of the text content for further in-depth analysis.

In view of the increasing frequency of people, especially young people, using emoticons such as emojis in text, and the increasingly close relationship between emojis and the emotional tendency of short-text content such as tweets, this paper aims to improve the accuracy of sentiment analysis on text data, especially short-text data, and objectively judge the emotional tendency of short-text content such as tweets. Combined with the features of emojis in short texts, this paper designs and implements a sentiment classification method with the emoji vectorization algorithm based on the blending of emojis and short texts.

## 2. Materials and Methods

### 2.1. Data Source and Corpus Construction

The corpus is one of the basic resources of natural language processing [22]. Compared with traditional texts, short texts such as tweets are characterized by short-text content, rich emoticons, more noise, and unstructured language [23]. Taking the short-text data of tweets as an example, it has four distinct characteristics. First, concise language. Although each tweet is limited to 280 words, most users often use only one or two sentences to express their views and opinions, and the number of words is far less than 280 words [24]. In addition, another important reason for concise language is that users often omit sentence components [25]. Short texts such as short tweets resulting in insufficient contextual information and difficult-to-extract evaluation objects due to default sentence elements have brought challenges to sentiment analysis. Second, various forms of expression. Emoticons are widely used in short texts such as tweets. According to the collected tweet data, it is found that the number of tweets containing at least one emoticon accounts for 37.5% of the total, which is enough to show the users’ love for emoticons. The reason is not only that emoticons increase the readability and sense of substitution of short text, but also because emoticons can directly and vividly convey users’ attitudes and emotions [26]. Third, more noise. Short text, such as tweets, uses specific symbols to indicate a specific role. Links often appear in tweets [27]. These symbols and links do not affect the emotional orientation of tweets and are the noise of sentiment analysis. If not removed, the accuracy of sentiment classification will be affected. Fourth, new words appear frequently on the Internet [28]. With the increasing number of netizens, netizens have created many new words which are different from traditional language forms in the process of online communication. For example, “TBH”, to be honest, and “amirite”, am I right, etc. Generally, network neologisms also have an emotional tendency. Therefore, in the field of tweet emotion analysis, we also need to analyze network neologisms.

This paper constructs a corpus of short texts containing rich emoticons. Data acquisition tools were used to collect short-text data using Twitter as the source of short texts, and 100,000 non-news tweets published from 15 October 2021 to 31 October, 2021, were selected as backup corpus. The preliminary collected tweet data contained a lot of noise and redundant data, so it was necessary to preprocess the data. First, we deleted tweets without emoticons. The research goal of this paper was to analyze the emotional features of the text by integrating emoticons with the short text. Emoticons can not only directly convey the feelings and opinions of the information publisher but also have an emotional tendency [29]. Therefore, emoticons should be considered as one of the important factors in the emotion analysis of short texts. In this paper, short-text and emoticon co-occurrence tweets are used as candidate corpus. Second, we deleted tweets with less than three words in the corpus. In general, we believe that tweets of less than three characters are not emotionally inclined or do not fully express the opinion holder’s attitude and should be removed from the corpus [30]. Third, we deleted links, usernames, topic names, and retweets from tweets in the corpus. Fourth, we performed word segmentation of Twitter text.

### 2.2. Annotation of Emotion in Short-Text Corpus

Short texts sent by users on social media can contain colorful emoticons. Among them, emojis are the most popular and most used emoticons. At present, emojis have been widely used in various social networks, and some of them have clear emotional tendencies. For example, 
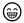
 has positive emotional tendencies, while 
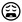
 has negative emotional tendencies. The site “Emojitracker” monitors emoji usage in tweets in real-time, as shown in Figure 1. The site ranks each emoji from highest to lowest in terms of the number of times they have been used in the current time. As can be seen from the figure, many of the emojis that are used frequently have no sentiment value, such as 
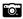
. These emojis have little impact on the sentiment orientation of tweets, so this paper includes them in the neutral category.

In addition, Cappallo et al. performed statistical analyses of a large amount of data with emoticons and found that the emergence of emoticons conforms to the long-tail distribution [31]. Maximum emojis can be covered by studying only the most frequent ones. Therefore, this paper adopts artificial methods to select the top 300 emojis with clear emotional tendencies from the 819 emoticons in the “Emojitracker” website for research. After screening, 80 emojis were selected for study in this paper, including 40 positive emojis and 40 negative emojis. Emojis and their emotional tendencies are shown in Table 1.

In this paper, two people were organized to annotate the selected short-text corpus data with emojis, and the content met the requirements by manual annotation. To ensure the accuracy of the annotation results, the candidate data could only be tweets with the same annotation results by two people. At the same time, a third person was selected to re-check the candidate data. Finally, a corpus of tweets emotion test was obtained by manual annotation, including 2000 positive and 2000 negative tweets each. All tweets in the corpus contain at least one emoji for analysis.

### 2.3. Emoji Vectorization Algorithm

#### 2.3.1. Word Vector Training

Emojis are strongly correlated with the sentiment orientation of short texts such as tweets. In the sentimental analysis of tweets, taking emojis as one of the research objects can more objectively judge the sentiment value of tweets. However, one of the questions that need to be addressed is how emojis in the form of pictures should be used to co-operate with the text to improve the accuracy of emotion classification. Eisner et al. proposed the vectorization algorithm called emoji2vec [32]. By transforming emojis into vector forms, emojis can be used in all areas of language processing as well as words.

Word2vec is a tool that Google open-sourced in 2013 to turn words in text into a data format that computers can understand [33]. It learns hidden information between words unsupervised in unlabeled training sets and obtains word vectors that can preserve syntactic and semantic relationships between words. The emoji2vec emoji vector algorithm uses the word2vec tool to train the word vector. The worf2vec tool was used for training in the processed corpus, and 765,285 five-dimensional word vectors were obtained after the training.

#### 2.3.2. Construct Sample Set

Emoticons and languages interact semantically. This paper constructs a sample set of emoticons according to the actual requirements of the vectorization algorithm. The sample set constructed consists of three parts: emoji, emoji name, and CLDR short name, such as: {
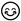
, Cute, Smiling Face with Smiling Eyes}.

The first component of the sample is the emoji picture. The emoji name is the second component of the sample. In the naming process, the emoji name must meet two conditions. First, the name can reflect the basic meaning of the emoji. Second, names are relatively unique [34]. The third component of the sample is CLDR. The Common Locale Data Repository (CLDR) project is a project of the Unicode Consortium to provide locale data in XML format for use in computer applications. CLDR contains locale-specific information that an operating system will typically provide to applications [35]. The Full Emoji List is a list of emojis that details the code and CLDR short name of each emoji. Sample information on emoticons can be found in this list. Through the above methods, we construct the positive samples in the sample set.

#### 2.3.3. Algorithm Flow

The emoji vectorization algorithm maps an emoji to a point in a high-dimensional vector space such that an emoji can be transferred into an N-dimensional vector with the format of (dim_1_, dim_2_, dim_3_, …, dim_N_). The algorithm steps are as follows.

Initialize the emoji vector $x_{i}$. Each sample contains the name of the emoji, and we take the word vector corresponding to the emoji name $w_{name}$ as the initial vector of the emoji vector. If the emoji name is an unregistered word, the emoji vector will be randomly initialized. The emoji vector $x_{i}=w_{name}$. It can be seen from the sample set information that the name of an emoji is a simple description of the meaning of the emoji, so the initial emoji vector already contains the basic part of the semantic information of the emoji, which will be conducive to the formation of the emoji vector.

Construct the description the vector $v_{j}$. $w_{1},\;w_{2},\;\ldots ,\;w_{N}$ is a set of word vector sequences, which, respectively, correspond to the word sequences in the descriptive sentences in the sample. In this paper, these word vectors are added together as description vectors of emoticons. Then, the formula to describe the vector is:(1)$$v_{j}=\overset{N}{\underset{k=1}{\sum }}w_{k}$$

Description vector is the sum of the corresponding word vectors of each word in a descriptive statement, which synthesizes the syntactic and semantic information of all words in a descriptive statement [36].

Establish the mathematical model. The dot product of emoji vector $x_{i}$ and description vector $v_{j}$ can indicate the similarity between the two vectors. The sigmoid function is used to model the similarity probability of emoji vector $x_{i}$ and description vector $v_{j}$, and the formula is:(2)$$P\left(y\right)=h{\left(x_{i}^{T}v_{j}\right)}^{y}{\left(1-h\left(x_{i}^{T}v_{j}\right)\right)}^{1-y}\;h\left(x\right)=\frac{1}{1+e^{-x}}$$

Calculate the emoji vector $x_{i}$. The sample dataset $D=\{\left(v_{j},\;y_{ij}\right)|v_{j}\in R_{n},\;y_{ij}\in \left\{0,\;1\right\}\}$ consists of every description vector $v_{j}$. When the description sentences j match with the emoji i, then $y_{ij}=1$. Otherwise, $y_{ij}=0$.

For all description vectors $v_{j}$ in the sample dataset D, the logarithmic loss function of Equation (2) is calculated, which is:(3)$$-\underset{i,j}{\sum }y_{ij}\log h\left(x_{i}^{T}v_{j}\right)-\underset{i,j}{\sum }\left(1-y_{ij}\right)\log(1-h\left(x_{i}^{T}v_{j}\right))$$

The batch gradient descent algorithm is used to find the best emoji vector $x_{i}$. The emoji vector obtained in this paper is a five-dimensional vector, and each emoji in the sample set has a corresponding emoji vector. Table 2 shows the vector of four emojis.

Visualization of five-dimensional emoji sentiment vectors in a two-dimensional space is displayed in Figure 2. Emojis include positive emojis and negative emojis.

### 2.4. Naïve Bayes

Naïve Bayes is a classification method based on Bayes’ theorem, which assumes conditional independence among features. When the naïve Bayes algorithm is applied to text classification, it assumes that the words above and below the text are independent of each other. The training set is counted and the prior probability of text category $C_{i}$ is calculated:(4)$$P\left(C_{i}\right)=\frac{N_{i}}{N}\;$$
where $N_{i}$ represents the total number of documents whose document category is $C_{i}$, and N represents the total number of all documents in the training set. Then, the conditional probability of the characteristic attributes of document d with classification is calculated:(5)$$P\left(d|C_{i}\right)=P\left(\left(t_{1},t_{2},t_{3},\ldots ,t_{n}\right)|C_{i}\right)=\overset{n}{\underset{j=1}{\prod }}P(t_{j}|C_{i})\;$$
where $t_{j}$ represents the j features of document d. $P(t_{j}|C_{i})$ represents the probability that feature $t_{j}$ appears in text category $C_{i}$. Finally, the formula for calculating the probability of all categories of documents to be classified is as follows:(6)$$P\left(C_{i}|d\right)=\frac{P\left(d|C_{i}\right)\cdot P\left(C_{i}\right)}{P\left(d\right)}\;$$

Therefore, document d is in the category with the highest probability. Naïve Bayes is a common text classification method with stable classification efficiency, can handle multiple classification tasks, and performs well on small-scale data.

### 2.5. Support Vector Machine

Support vector machine (SVM) is a kind of classifier whose core idea is to determine an optimal hyperplane that can correctly divide samples into two classes by maximizing the interval of the nearest samples in different classes of samples in the training set [37].

Given the i training sample in a sample set $\left(x^{\left(i\right)},y^{\left(i\right)}\right)$, where x represents the eigenvector, y={−1, 1} represents the class tag. When the linear is separable, the hyperplane can be expressed as:(7)$$w^{T}x+b=0\;$$

Hence, for any sample set $\left(x^{\left(i\right)},y^{\left(i\right)}\right)$, when $y^{\left(i\right)}=1$, $w^{T}x+b>0$, and when $y^{\left(i\right)}=-1$, $w^{T}x+b<0$. We define:(8)$$\{\begin{array}{c} w^{T}x+b\geq 0,\;y^{\left(i\right)}=1\\ w^{T}x+b\leq 0,y^{\left(i\right)}=-1 \end{array}$$

The sum of the distances between the two support vectors belonging to different categories and the hyperplane is:(9)$$\gamma =\frac{2}{\left\|w\right\|}\;$$

In order to determine the optimal hyperplane, it is necessary to satisfy the parameters w and b in Equation (8), such that the interval γ is the largest, namely:(10)$$\{\begin{array}{c} min\frac{1}{2}{\left\|w\right\|}^{2}\\ y^{\left(i\right)}\left(w^{T}x^{\left(i\right)}+b\right)\geq 1,i=1,2,3,\ldots ,n \end{array}\;$$

In this paper, the support vector machine (SVM) algorithm, which is widely used in the classification field, is selected as the classification algorithm to construct an SVM emotion classifier. It is crucial to select suitable features to get a better SVM emotion classifier. According to the characteristics of short texts, this paper selects the following text features.

First, the frequency of emotional words. Counting the frequency of emotion words requires an emotion dictionary. The emotion dictionary used in the experiment is Word-Emotion Association Lexicon [38]. Each short text in the experimental data is traversed, and the number of positive emotion words and negative emotion words is counted according to the emotion dictionary. Due to the different lengths of each short text, normalization is needed. The number of emotional words obtained by statistics is divided by the total number of words in the short text to obtain the frequency of emotional words.

Second, negative words and adverbs of degree. When people communicate, they habitually use negative words and adverbs of degree. Although negative words and adverbs of degree do not have emotional polarity, when they are combined with emotional phrases, they will affect the original emotional tendency of emotional words [39]. Specifically, the combination of degree adverb plus emotional words will enhance or weaken the original emotional tendency of emotional words, such as “really fancy”. The combination of negative words plus emotion words will reverse the polarity of emotion words, such as “not into”. The combination of degree adverb plus negative word plus emotion word will make the emotion degree and polarity of emotion word change, such as “really not into”. Therefore, special attention should be paid to negative words and adverbs of degree when analyzing the emotional tendency of short texts.

Third, the number of exclamation marks and question marks. In short-text content, there are often multiple question marks or multiple exclamation marks used together. The combination of punctuation marks indicates the strengthening of the original emotional tendency. For example, the combination of multiple exclamation marks indicates the strengthening of surprise, anger, and other emotions. The combination of multiple question marks indicates the strengthening of doubts and puzzles. Therefore, counting the number of exclamation marks and question marks in short-text content is helpful to analyze the emotional tendency of short texts.

Fourth, the number of emoticons such as emojis. Emoticons have their own emotional tendency, which affects the whole process of short text to a certain extent. The emotional intensity of the body is even the emotional tendency. Therefore, the number of emoticons is one of the characteristics of this paper.

### 2.6. Convolutional Neural Network

Deep learning has been widely used in image recognition, speech recognition, computer vision, and other fields since it was proposed and has made remarkable achievements [40]. Compared with traditional machine learning algorithms, deep learning has advantages in feature expression and model building [41]. Therefore, we use the convolutional neural network (CNN) to analyze short-text emotion. To give full play to the role of emoticons in promoting short-text emotional tendency, this paper adds emoticons vector to the short-text emotional analysis based on the CNN classification model.

The convolution layer and pooling layer play an important role in the convolution neural network. The convolution layer can extract local features and semantic combinations from input data. The pooling layer selects local features and semantic combinations based on the convolution layer and then filters out unimportant local features and semantic combinations with low confidence [42]. The alternating superposition of multiple convolution layers and pooling layers can extract highly abstract features from text data and improve the accuracy of emotion classification. Figure 3 is the structure diagram of the classification model based on the convolutional neural network adopted in this paper.

The model has the following four layers:

Input layer. The input of the classification model is a matrix. The matrix is formed by connecting the word vectors corresponding to all words in the sentence after word segmentation. If the word vector corresponding to the ith word in the input sentence with length n is $X_{i}\in R^{5}$, then the matrix is $X=X_{1}⊕X_{2}⊕\ldots ⊕X_{n}$, and ⊕ is the connector.

Convolutional layer. The classification model based on the convolution neural network uses convolution filters with different window h lengths to extract the local features of the input layer. In the research, we implement the parallel convolution layer with multiple convolution kernels of different sizes to learn short-text features. Multiple convolution kernels are defined to acquire features in the short-text content and reduce the degree of fortuity in the feature extraction process. We define the filter size of convolution kernels as $h_{1}Xk$, $h_{2}Xk$, $h_{3}Xk$, where k is an integer and the dimension of word embeddings, and $h_{1}$ is the stride value. The feature obtained by using the convolution filter w as the input layer is:(11)$$c_{i}=f\left(w\times X_{i:i+h-1}+b\right)\;$$
where f is the non-linear activation function. The rectified linear unit (ReLu) is used in the research:(12)$$c_{i}=max\left(0,w\times X_{i:i+h-1}+b\right)\;$$

In the equation, b is the bias term, $w\in R^{hk}$ is the shared weight, $X_{i:i+h-1}$ represents the connection of the word embedding which is from the i word of short text X to the i+h−1 word ordered from top to bottom, and b∈R is an offset term. A characteristic graph can be obtained by applying the convolution filter to all adjacent word vectors with length *h* in the input matrix $C=\left[c_{1},c_{2},\;\ldots ,c_{n-h+1}\right]$, $c\in R^{n-h+1}.\;$Therefore, n−h+1 feature maps are used for each convolution kernel to obtain a feature vector t whose dimension is X (n−h+1).

If the number of convolution kernels is p, then p feature vectors can be obtained through feature mapping, and $T=\left[t_{1},\;t_{1},\ldots \ldots ,t_{p}\right]$. If q parallel convolution kernels of different types are used; for example, $h_{1}Xk,\;h_{2}Xk,\;\ldots \ldots ,\;h_{q}Xk$, and the number of each type of convolution kernel is p, then pXq feature vectors that can be obtained after feature mapping, and $S=\left[t_{1},\;t_{1},\ldots \ldots ,t_{p}\right]$. Therefore, S is the output from the convolution layer. The output will be sent to the pooling layer of the CNN.

Pooling layer. The role of the pooling layer is to screen out the optimal local features. The pooling layer performs a max-pooling operation on all the characteristic graphs obtained by the convolution layer, which is $\hat{C}=\max\left\{C\right\}$.

Fully connected layer. The output of the pooling layer is connected to the output node of the last layer by full connection, and the tweet emotion is classified by the SoftMax classifier. In the final implementation, the dropout technique is used on the fully connected layer to prevent the hidden layer neurons from self-adapting and to reduce overfitting. The output layer is a fully connected SoftMax layer with the dropout technique. The output layer outputs the classification accuracy and loss of the method. The proportion of dropout starts from 0.5 and gradually decreases until the model performs best, which is 0.2. The number of training epochs is 8. The optimizer used for training is AdamOptimizer.

### 2.7. Recurrent Neural Network

The recurrent neural network (RNN) is one of the artificial neural networks. It is a neural network that can model sequence data and process sequence data of any length. The difference between it and the convolutional neural network (CNN) is that the cyclic neural network can consider the sequence characteristics in the text [43]. In traditional neural networks, nodes in the same layer are not connected. Traditional neural networks assume that elements do not affect each other. However, this assumption does not accord with the realistic logic. Therefore, this network structure will appear powerless in dealing with many problems. The nodes between the hidden layers of the recurrent neural network are connected. The input of the current time and the output of the hidden layer of the previous time jointly determine the output of the current time. In other words, the recurrent neural network can remember the previous information. This sequence characteristic is that when RNN processes the current input information, it will calculate the current text together with the previously memorized information.

The RNN includes the input layer, the hidden layer, and the output layer, as displayed in Figure 4 [44]. It can be clearly seen from the figure that the nodes of the hidden layer can not only be self-connected but also interconnected.

In the RNN, the hyperbolic tangent activation function tanh is used to determine the output of the current network unit and transfer the current network unit state to the next network unit:(13)tanh(x)=2σ(2x)−1 
(14)$$f\left(x\right)=\frac{1-e^{-2x}}{1+e^{-2x}}$$

### 2.8. Long Short-Term Memory

The backpropagation through time (BPTT) algorithm is used in the training of the RNN. In the training process of this algorithm, there will be the problems of gradient explosion and gradient disappearance, which makes the RNN unable to deal with long sequences. The long short-term memory (LSTM) network is an improvement of cyclic neural networks and solves the above problems [45]. LSTM solves the problem of gradient explosion encountered in the RNN, and it is accurate to extract text long-distance dependent semantic features when processing text information [46]. LSTM solves the problem of text long-distance dependence through the gating system in the network unit. LSTM contains three such gating systems, namely input gate, output gate, and forgetting gate. These gating systems are realized by the sigmoid function:(15)$$\sigma \left(x\right):\;f\left(x\right)=\frac{1}{1+e^{-x}}$$

The output of the sigmoid function is a value between 0 and 1. The closer the value is to 1, the more information the door opens and retains. On the contrary, the closer it is to 0, the more information it needs to forget. The single-cell unit of LSTM is composed of three sigmoid functions, two tanh activation functions and a series of operations.

The structure of the diagram LSTM cell is displayed in Figure 5 [47] below.

In the LSTM neural network, the first step is to process the current input information and the information transmitted from the previous state through the forgetting gate to determine which information will be lost from the cell state. The second step is to use the input gate to control the input of useful information into the current state and obtain the latest state of the current state through the tanh activation function. The output determines the state of the output and passes it to the next door.

## 3. Results

We introduced the convolutional neural network in deep learning to extract hidden sentiment features from short-text data and find the best analysis method for the fusion of short-text content and emoticons by comparing different classification methods.

Based on the convolutional neural network classification model, this paper analyzes three classification models. The first classification model only considers short texts in the tweet corpus, removes emoticons, divides each tweet in the experimental data into words, and takes the sentence matrix connected by the corresponding word vector of all words as the input of the convolutional neural network classification model to classify short texts. The second classification model first converts emoticons in the short-text corpus into named texts corresponding to emoticons. For example, it would convert the tweet: “Getting everybody together for the start of the Christmas tour! 
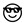
” into “Getting everybody together for the start of the Christmas tour! Smiling face with sunglasses”. Then, the transformed tweets are segmented into sentence matrices, which are trained and tested by the classification model of the convolutional neural network. The third classification model is to transform emoticons into emoticons vectors using the emoji vectorization algorithm, and then connect the corresponding word vector and emoticons vector into the sentence matrix according to the lexical order of the tweet corpus. Finally, the sentence matrix is input into the convolutional neural network classification model for classification.

We conducted comparative experiments on the previously established corpus of tweet data and used naïve Bayes, LSTM, RNN, and SVM as the baseline methods. In the experiments, the Python programming language and the TensorFlow platform were utilized for implementation. The experimental results of the corpus with positive sentiment value and negative sentiment value are shown in Table 3 and Table 4.

According to the comparative analysis of experimental results, it can be concluded that the model based on deep learning performs better, namely in accuracy, because the deep learning method can extract deep-seated data features in short texts such as tweets. In the short-text data set containing rich emoticons, the best experimental model is the third one, which converts emoticons and text into vectors and analyzes them. By comparing the experimental outcomes of the analysis of positive emotions and negative emotions, we can find that the accuracy of all models on the analysis of negative short-text emotions is reduced, especially the second model. When emoticons are converted into text messages and then combined with short-text content analysis, the reduction range is the largest, and the instability is the highest. This is because many short texts with negative emoticons may not express negative emotions but express things such as surprise, emotion, etc. In this case, the conversion of emoticons into words to analyze the emotional tendencies of the text will have the opposite effect. In general, converting emoticons and texts into vectors to achieve the highest accuracy in analyzing emotional value, which reflects that emoticons vectorization algorithm can play a significant role in the emotional analysis of short texts.

Novak et al. provide a mapping to positive, negative, and neutral occurrence information for 751 emojis, also available on Kaggle [48]. Based on the dataset with 70,000 tweets and 969 different emojis, we designed a contrast experiment. We first extract the tweets from the dataset that contained the emojis in Table 1. A total of 37,810 tweets were selected from the dataset. Then, we proceeded with the accuracy analysis experiment with different methods based on the corpus with extracted data.

The evaluation indices of the experiment included accuracy (P), recall (R), positive and negative class F1 values. For the overall performance, the overall correction rate accuracy is used, and the calculation formula is:(16)$$Accuracy=\frac{TP+TN}{TP+FP+TN+FN}$$

In this formula, *TP*, *FP*, *TN*, and *FN* represent the correctly classified positive tweets, the misclassified positive tweets, the correctly classified negative tweets, and the misclassified negative tweets, respectively.

The experiment compared the proposed method CNN (Emoji2Vec, Word2Vec) with traditional classification methods, including naïve Bayes, CNN (Word2Vec), CNN (Emoji2Word, Word2Vec), LTSM, RNN, and SVM. The parameter setup is as follows. The parameters of c and g of SVM are obtained by grid search. The parameter setup of naïve Bayes is the default parameter setup of sklearn. RNN is MV-RNN of reference, and its parameter setup is the same. LSTM is Tree-LSTM of reference, and its parameter setup is the same. The emotional classification results are displayed in Table 5.

The experimental results indicate that CNN (Emoji2Vec, Word2Vec) has a better performance with higher overall accuracy and two types of F1 values than the traditional classification methods. The naïve Bayes algorithm has the lowest overall performance indices. The experimental results show that the CNN (Emoji2Vec, Word2Vec) method is effective for the emotion classification of short texts with emoticons such as emojis.

## 4. Discussion

Entropy refers to the degree of the chaos of a system. A system with a low degree of chaos has low entropy, while a system with a high degree of chaos has high entropy [49]. In the absence of external interference, entropy increases automatically [50]. In information theory, entropy is the average amount of information contained in each piece of information received, which is also called information entropy [51]. In the information world, the higher the entropy, the more information can be transmitted, and the lower the entropy, the less information can be transmitted [52]. The booming development of social media such as Twitter and TikTok with the increasingly wide range of short-text communication has reduced the difficulty of information dissemination. At the same time, the extensive use of emoticons such as emojis in short texts increases the amount of information covered in short-text content and boosts the entropy value of short-text information, which makes the prediction of short-text information content represented by sentiment characters difficult.

The topic of this paper is to improve the accuracy of identifying sentiment features of short texts with emoticons, such as emojis. Based on this research goal, we first established a corpus containing rich emoticons and short texts and identified their sentiment tendencies in an artificial way, which are used as a data source for subsequent analysis. Second, we screened the emojis commonly used in 819 social media and selected 40 emojis with positive emotions and 40 emojis with negative emotions, respectively. Third, we built an algorithm to convert emoticons into vector information and analyzed the emojis we selected. Fourth, we combined the vector information transformed by emojis with the vector information transformed by characters in the short-text content and analyzed the sentiment tendency of the short-text content by using the CNN model. The results were compared with simple text analysis and emoticon conversion, and the proposed method improved the accuracy of identifying positive and negative emotional tendencies.

Existing analysis methods of short-text emotion tendency adopt analysis methods such as combining with context. For example, Wan et al. proposed an ensemble sentiment classification system of Twitter data [53]. Although they accurately analyzed the sentimental characteristics of the content of the short text, they removed the punctuation, symbols, emoticons, and all other non-alphabet characters from the short text and hence ignored the important factor of emoticons in the emotional tendency of the short text. Some studies have analyzed the emotional value of emoticons. For example, Mohammad et al. designed an algorithm and method for sentiment analysis using texts and emoticons [54]. Matsumoto et al. developed an emotion estimation method based on emoticon image features and distributed representations of sentences [55]. However, nowadays, the majority of people use emojis to express their sentiment in short texts, and emojis dominate the use of emoticons. The above-mentioned methods only focus on emoticon symbolic expression tokens and text-based emoticons, which face difficulties in analyzing short texts with emojis. Therefore, based on the existing text vectorization algorithm and emoticon vectorization algorithm, this paper designs a sentiment classification method that integrates emoticons and characters. This method has been proved to be effective. Compared with analyzing the text content of short texts or emoticons of short texts, the method proposed in this paper has higher sentiment character recognition accuracy.

Sentiment feature analysis can help organizations and enterprises collect and analyze users’ attitudes towards their products or services in public opinion and help improve them, which is an effective means to improve the efficiency of data analysis. An increasing number of systems and methods are designed and applied for this. Our method, which blends short texts with emoticons, still has room for improvement in its accuracy in identifying negative sentiment tendencies. The analysis shows that emoticons with negative emotions contain more complex emotional information than emoticons with positive emotions. Not only can it express negative emotions, but it can also express emotions such as movement, surprise, and even pleasantness. Therefore, our future research direction and improvement is to improve the efficiency and accuracy of methods to identify texts with negative emotions.

## 5. Conclusions

Analyzing and understanding the sentimental characteristics of short texts is conducive to obtaining the data information of the deep value of public opinion and helping organizations and companies optimize their services and products. Airline services have been improved and enhanced after implementing the ensemble sentiment classification method [53]. Fast-moving consumer goods (FMCG) brands such as P&G and hotels such as Marriott Corporation mine and analyze consumers’ sentiment opinions from short texts in social media [56,57]. These implementations effectively improve the production efficiency of enterprises and boost users’ satisfaction, which achieves a win–win situation. We advocate further study in this research field.

In this paper, we propose a short-text sentiment analysis method that combines emoticons such as emojis with short-text content. A great number of tweets containing emojis are processed and analyzed to obtain the sentimental characteristics of short texts such as tweets. This paper first classifies popular emoticons, converts emoticons together with characters into vector representations, and analyzes them using the convolutional neural network method. Experimental results show that the proposed method is more accurate than the existing method.

## Figures and Tables

**Figure 1 entropy-24-00398-f001:**
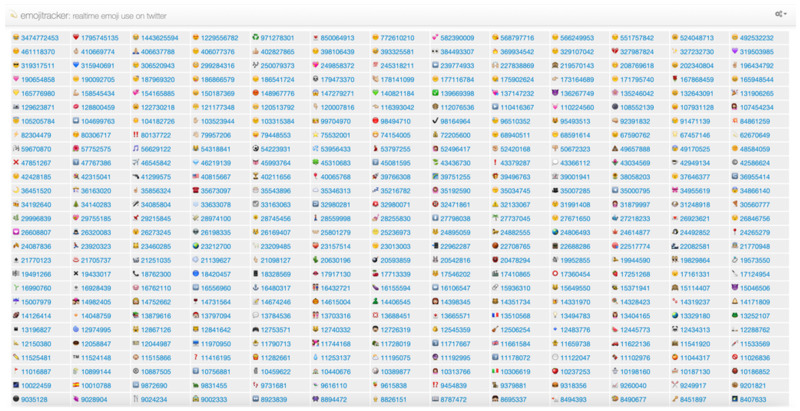
The site “Emojitracker” monitors emoji usage in tweets in real time.

**Figure 2 entropy-24-00398-f002:**
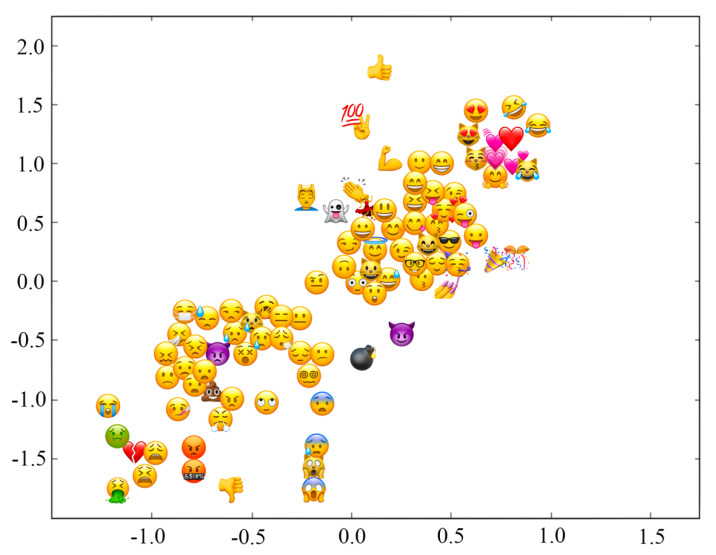
Visualization of five-dimensional emoji sentiment vectors in a two-dimensional space.

**Figure 3 entropy-24-00398-f003:**
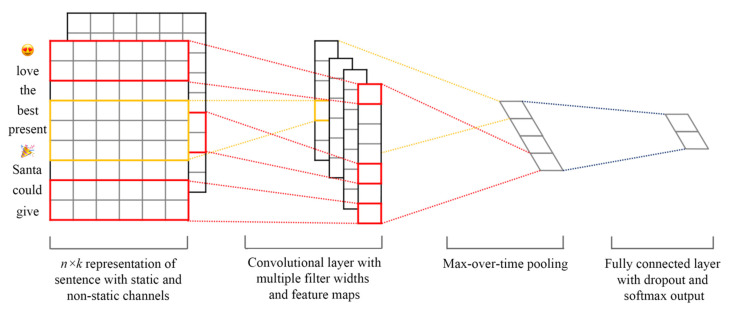
Structure diagram of the classification model.

**Figure 4 entropy-24-00398-f004:**
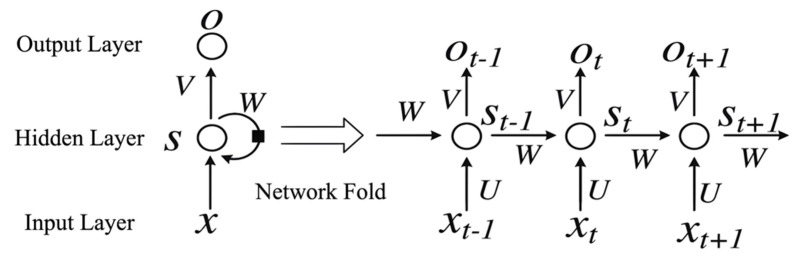
Structure of RNN.

**Figure 5 entropy-24-00398-f005:**
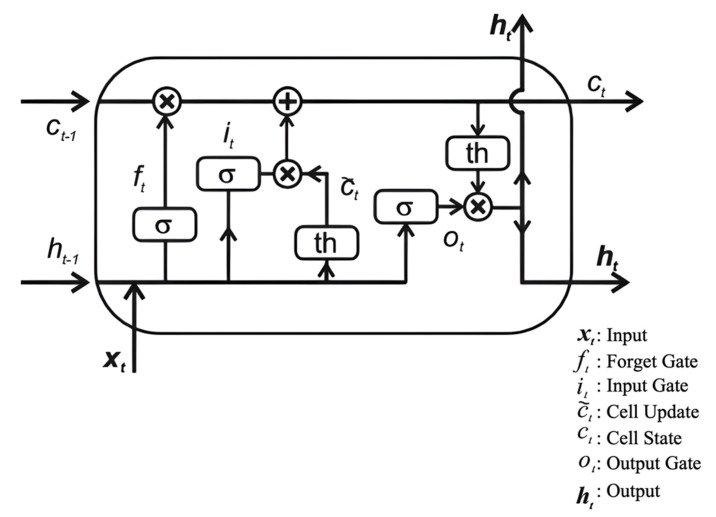
Structure diagram of LSTM cell.

**Table 1 entropy-24-00398-t001:** Emojis and their emotional tendencies.

Emotional Tendencies	Emojis
Positive	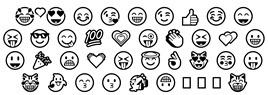
Negative	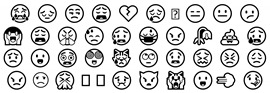

**Table 2 entropy-24-00398-t002:** Vectors of four emojis.

Emoji	Emoji Vector
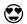	1.253765409217565−0.9265875698769671.698378103032509−0.5278348923464270.847561023874692
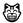	1.157409314569509−0.3750257908461350.746948348694133−1.0478915346759270.287905347982658
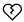	−2.2192470913470050.345388574098972−1.6137829457826910.547893128730935−0.789132897543139
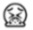	−0.4417089353870971.134897523487950−1.8245608239541660.078318930571228−0.927943898316451

**Table 3 entropy-24-00398-t003:** Experimental results with positive-sentiment-value corpus.

Analysis Method	Identify Quantity	Accuracy
Naïve Bayes	1447	72.35%
CNN (Word2Vec)	1632	81.60%
CNN (Emoji2Word, Word2Vec)	1649	82.45%
CNN (Emoji2Vec, Word2Vec)	1704	85.15%
LSTM	1660	83.00%
RNN	1651	82.55%
SVM	1558	77.90%

**Table 4 entropy-24-00398-t004:** Experimental results with negative-sentiment-value corpus.

Analysis Method	Identify Quantity	Accuracy
Naïve Bayes	1351	67.55%
CNN (Word2Vec)	1473	73.65%
CNN (Emoji2Word, Word2Vec)	1385	69.25%
CNN (Emoji2Vec, Word2Vec)	1596	79.80%
LSTM	1479	73.95%
RNN	1470	73.50%
SVM	1402	70.10%

**Table 5 entropy-24-00398-t005:** Comparison of emotional classification results.

**Analysis Method**	**Positive P**	**Negative P**	**Positive R**	**Negative R**	**Positive F1**	**Negative F1**	**Accuracy**
Naïve Bayes	79.15%	82.58%	82.37%	78.66%	80.27%	79.36%	80.75%
CNN (Word2Vec)	86.38%	87.75%	87.20%	85.52%	87.05%	86.97%	86.91%
CNN (Emoji2Word, Word2Vec)	85.60%	89.95%	87.42%	85.10%	87.83%	86.65%	87.60%
CNN (Emoji2Vec, Word2Vec)	89.23%	91.60%	92.16%	88.33%	90.59%	89.70%	90.35%
LSTM	86.55%	87.10%	87.05%	88.42%	86.98%	88.15%	87.20%
RNN	84.92%	86.03%	87.13%	84.88%	86.04%	84.79%	85.33%
SVM	83.78%	85.35%	85.41%	83.88%	84.79%	84.32%	84.66%

## Data Availability

Not applicable.

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
