# Peer review of "Sentiment Classification Method Based on Blending of Emoticons and Short Texts"

_entropy, 2022, doi:10.3390/e24030398_

Round 1
Reviewer 1 Report
The paper describes a sentiment classification based on short text and emoticons. The results are used also with a CNN and appears to outperform on this dataset. The dataset is described with edits though results are difficult to reproduce if it is not available for others to test. I would recommend adding this. Also the ground truth is not clear to me how it is determined. So this should be clarified in the paper.
Author Response
Response to Reviewer 1 Comments
Point 1:
The paper describes a sentiment classification based on short text and emoticons. The results are used also with a CNN and appears to outperform on this dataset. The dataset is described with edits though results are difficult to reproduce if it is not available for others to test. I would recommend adding this.
Response 1: We add an experiment on the Kaggle tweets database in the Result section. The dataset is already classified according to the sentiment value by the developers Novak et al. We implemented our method on this database for accuracy check. Others have access to the Kaggle dataset and test the results proposed in the manuscript.
Point 2: Also the ground truth is not clear to me how it is determined. So this should be clarified in the paper.
Response 2: The calculation formula of accuracy is added and displayed in the Result section. Also, the Discussion section has been improved.

Reviewer 2 Report
Brief summary
The paper proposes a CNN-based method for sentiment classification of short texts containing emoticons. Text and emoticons are converted into numerical vectors and used as input to the CNN. Results show that including emoticons as input to the classifier improves classification accuracy from 81.6% (only text) to 85.15% (emoticon + text).
General Comments
In the introduction, the authors mention as their main novel contribution the inputting text and emoticons together to a CNN for sentiment classification. However, this has been done before see
-
Mohammad Aman Ullah et al, An algorithm and method for sentiment analysis using the text and emoticon, ICT Express, Volume 6, Issue 4, 2020, Pages 357-360, ISSN 2405-9595, https://doi.org/10.1016/j.icte.2020.07.003.
-
Wan Y., Gao Q. An ensemble sentiment classification system of twitter data for airline services analysis 2015 IEEE International Conference on Data Mining Workshop (ICDMW) (2015), pp. 1318-1325.
-
Fujisawa, A.; Matsumoto, K.; Yoshida, M.; Kita, K. Emotion Estimation Method Based on Emoticon Image Features and Distributed Representations of Sentences. Appl. Sci. 2022, 12, 1256. https://doi.org/10.3390/app12031256
Furthermore, Ullah et al have also used CNN and several other machine learning methods, and by using LSTM RNN they obtained an accuracy of 0.89 which is higher than the one reported in this work.
In terms of the evaluation of their method, the CNN architecture proposed should be compared with state of the art methods for sentiment classification and that would be RNN-based methods. Thus, the comparison with Naive Bayes is not sufficient to show that the proposed method is better at this task than other approaches. Additionally, authors should present several performance metrics such as AUROC, and F1 in addition to accuracy.
In terms of reproducibility, the methods section does not give enough details to allow someone to repeat their experiment. For example, the data should be made available, the details of the CNN architecture used should also be provided: what activation function is being used? How many units are per layer? What is the size of the kernel/filters in the convolutional layers? What is the percentage of dropout used? What optimizer was used for training? How many training epochs? What language and libraries were used for implementation? Tensorflow? Keras? Pytorch? Additionally, I would recommend to use a data set that has been used before so that it facilitates comparison between approaches (kaggle has a tweet data set).
Specific comments
-
Figure 1 is unreadable (too small)
-
There are several typos in the text
-
Suggest to remove Table 2. Figure 2 is great to illustrate emoticon vectors and their similarity.
-
The entropy discussion in Discussion section is unrelated to the topic of the article.
Author Response
Response to Reviewer 2 Comments
Point 1:
In the introduction, the authors mention as their main novel contribution the inputting text and emoticons together to a CNN for sentiment classification. However, this has been done before see:
Mohammad Aman Ullah et al, An algorithm and method for sentiment analysis using the text and emoticon; Wan Y., Gao Q. An ensemble sentiment classification system of twitter data for airline services analysis; Fujisawa, A.; Matsumoto, K.; Yoshida, M.; Kita, K. Emotion Estimation Method Based on Emoticon Image Features and Distributed Representations of Sentences.
Response 1:
Short text sentiment value analysis is a popular topic in the field of machine learning and natural language analysis and it has been discussed for years. In this paper, instead of studying the sentiment classification of short text, we focus on the following parts: First, the boost in usage of emoticon, specifically, emoji in short text in recent two years. Second, how the emoji dominate the emotional orientation. Third, what method should we design and use to analyze the sentiment value of short text with emoji.
From the paper of Mohammad Aman Ullah et al, An algorithm and method for sentiment analysis using the text and emoticon, although they analyzed the sentiment values of text and emoticon, they did not focus on short text with emoji. In these two years, the 5G era, the majority of short texts nowadays contain emojis such as ? and ??, people use emojis to express their sentiment instead of using the emoticons symbolic expressions tokens, the products of 2G era, such as \:'', \='', \-'', \)'' discussed in the paper of Mohammad Aman Ullah et al. Although people may use emoticons symbolic expressions tokens to express their feelings nowadays, emojis have gone mainstream. As a result, the method proposed by Mohammad Aman Ullah et al faces difficulties in analyzing short texts with emojis.
From the paper of Wan Y., Gao Q. An ensemble sentiment classification system of Twitter data for airline services analysis, they remove the punctuations, symbols, emoticons and all other non-alphabet characters from the short text, which means they did not analyze short texts with emojis.
From the paper of Fujisawa, A.; Matsumoto, K.; Yoshida, M.; Kita, K. Emotion Estimation Method Based on Emoticon Image Features and Distributed Representations of Sentences. Same as the first paper, they mainly focus on analyzing the text-based emoticons such as (*-!-)b in the short text. As a result, their method faces difficulties in analyzing short texts with emojis.
Although the above-mentioned methods face difficulties in analyzing the sentiment value of short texts with emojis, we will add them to the Conclusion and Discussion parts for comparing the differences between this paper and other results, and expanding also enriching the content of the manuscript.
Point 2:
Furthermore, Ullah et al have also used CNN and several other machine learning methods, and by using LSTM RNN they obtained an accuracy of 0.89 which is higher than the one reported in this work.
Response 2:
Mohammad Aman Ullah et al use machine learning methods to analyze the short texts with emoticons symbolic expressions tokens. These text-based emoticons are relatively more direct in sentiments and opinions and contain relatively simpler information than emojis. Compared with text-based emoticons, emojis are more complex, manifold, and contain diverse emotional information, especially in the negative sentiment values which are more difficult to analyze. Therefore, it is understandable that Ullah et al obtained a higher accuracy because the emoticons in the short texts in their research are more direct in reflecting the sentiment information. What’s more, the corpus for tests and experiments in both papers is also different. In our research, all short texts contain at least one emoji for analysis, which is another reason for an affordable lower accuracy.
We add results comparation between methods proposed in the paper of Mohammad Aman Ullah et al and the methods we designed in Results part. We also add comparation between methods proposed in other research and methods proposed in this paper base on one kaggle databas. We expand the discussion with information and results from the the paper of Mohammad Aman Ullah et al.
Point 3:
In terms of the evaluation of their method, the CNN architecture proposed should be compared with state of the art methods for sentiment classification and that would be RNN-based methods. Thus, the comparison with Naive Bayes is not sufficient to show that the proposed method is better at this task than other approaches. Additionally, authors should present several performance metrics such as AUROC, and F1 in addition to accuracy.
Response 3:
We add information about RNN and LSTM.
We add comparations of results between methods proposed in this paper with RNN-based methods, LSTM based methods.
We add performance metrics F1 to accuracy in the Result part.
Point 4:
In terms of reproducibility, the methods section does not give enough details to allow someone to repeat their experiment. For example, the data should be made available, the details of the CNN architecture used should also be provided: what activation function is being used? How many units are per layer? What is the size of the kernel/filters in the convolutional layers? What is the percentage of dropout used? What optimizer was used for training? How many training epochs? What language and libraries were used for implementation? Tensorflow? Keras? Pytorch? Additionally, I would recommend to use a data set that has been used before so that it facilitates comparison between approaches (kaggle has a tweet data set).
Response 4:
We add detailed information on the CNN architecture:
Activation function has been added in equation 12.
Number of units in each layer has been added.
Size of the kernel/filters in the convolutional layers has been added.
The percentage of dropout is added.
The optimizer used for training is added.
The number of training epochs is added.
Language and libraries used for implementation added in the Result part.
We add the kaggle tweet dataset in the experiment.
Point 5:
Figure 1 is unreadable (too small).
Response 5:
We enlarge Figure 1 for reading.
Point 6:
There are several typos in the text.
Response 6:
Typos have been corrected.
Point 7:
Suggest to remove Table 2. Figure 2 is great to illustrate emoticon vectors and their similarity.
Response 7:
We create the table to display the results of the emoji sentiment vectors. It’s a five-dimensional vector and to visualize the results directly, we propose to convert the five-dimensional vectors results into a two-dimensional space. However, the transition may loss accuracy. So we would like to keep the Table 2 for showing the results of the vectoriazation algorithm.
Point 8:
The entropy discussion in Discussion section is unrelated to the topic of the article.
Response 8:
The main idea for the entropy discussion in the Discussion section is we want to reveal the relationship between the conclusion of this manuscript and the information entropy and how the research outcomes improve the information entropy field. We also want to strengthen the connection between the research of this manuscript and the journal because of the name Entropy

Reviewer 3 Report
The article Sentiment Classification Method Based on Blending of Emoticons and Short Texts was prepared using secondary data. The paper was correctly submitted for review. The method of blending and using emoticons in short text messages was presented in an interesting way. I rate the literature review and the research conducted well. The discussion is also correctly conducted.
Conclusions should be expanded, there is no reference to the literature review in the conclusions, what are the implications? Or will the authors advocate further research on the issue of emoticons?
The last sentence "The experimental results show that the proposed method is more accurate than the existing method" is very general - it should be expanded, which existing method? specifically indicate.
Author Response
Response to Reviewer 3 Comments
Point 1:
Conclusions should be expanded, there is no reference to the literature review in the conclusions, what are the implications? Or will the authors advocate further research on the issue of emoticons?
Response 1:
Content about reference, implications, and the author's attitude towards the research field has been added in the Conclusion section.
Point 2:
The last sentence "The experimental results show that the proposed method is more accurate than the existing method" is very general - it should be expanded, which existing method? specifically indicate.
Response 2:
Comparison of existing analysis methods and the method proposed in this paper is discussed in the Introduction and Discussion sections, detailed information of the comparison has been added.

Round 2
Reviewer 2 Report
Thank you for all the changes. I think the manuscript has improved considerably.
Two main comments:
- Figures 4 and 5 were probably taken from a textbook so references should be added.
- The code and data should be made available in a repository such as GitHub or Zenodo
Author Response
Point 1:
Figures 4 and 5 were probably taken from a textbook so references should be added.
Response 1:
References have been added.
Point 2:
The code and data should be made available in a repository such as GitHub or Zenodo
Response 2:
Relevant code, data, model, and projects have been updated on the GitHub repository: zouhaochen/Short_Text_Sentiment_Analysis and are now available by the public. Access will be also added to the Supplementary Materials, Data Deposit and Software Source Code Part of the submission.
